# HMGB1 and Its Signaling Pathway in Osteosarcoma: Current Advances in Targeted Therapy

**DOI:** 10.3390/cimb47110887

**Published:** 2025-10-27

**Authors:** Zhuosheng Liu, Fucai Wang, Zhihan Zhou, Mei Wu, Qinghua Huang, Xinpeng Jiang, Xuan Wen, Liuting Ye

**Affiliations:** School of Medicine, Huaqiao University, Quanzhou 362021, China

**Keywords:** high-mobility group box 1, osteosarcoma, signaling pathway, targeted therapy

## Abstract

This article reviews the research progress for high-mobility group protein B1 (HMGB1) and its signaling pathway in osteosarcoma (OS) and discusses its application potential in targeted therapy. A large number of domestic and foreign studies were reviewed to summarize the research results on the the biological function, signal pathway regulation mechanism, and intervention strategy of HMGB1 in recent years. HMGB1 promotes OS cell proliferation, invasion, and immune escape by activating RAGE, TLR4, and downstream MAPK, NF-κB, and PI3K/AKT signaling pathways. Interfering with HMGB1 or its signaling axis shows good antitumor potential in in vitro and in vivo models, but clinical transformation is still limited by its dual biological effects and tumor heterogeneity. HMGB1 and its related signaling pathways are important targets for the treatment of osteosarcoma. In the future, the development of a multi-channel combined intervention and efficient delivery system will provide a new direction for improving the therapeutic effect.

## 1. Introduction

As the most common primary malignant bone tumor, osteosarcoma (OS) poses a serious threat to the lives and health of adolescent and elderly patients due to its highly aggressive nature, propensity for early metastasis, and resistance to treatment [1,2]. Although the application of multimodal comprehensive treatment strategies (including neoadjuvant chemotherapy, radical surgery, and adjuvant radiotherapy) has improved clinical efficacy, the five-year survival rate of metastatic or recurrent patients has not been significantly improved, and this therapeutic bottleneck urgently requires a breakthrough through novel molecularly targeted intervention strategies [3,4].

With the development of molecular oncology, the key role of high-mobility group box 1 (HMGB1), a multifunctional non-histone molecule, in developing OS and drug resistance has been gradually revealed [5,6]. HMGB1 has unique biphasic localization characteristics: in its physiological state, it is mainly located in the nucleus, participating in DNA damage repair and chromatin remodeling, while in cellular stress, necrosis, or inflammation, it can be actively secreted or passively released into the extracellular region, where it can activate key signaling pathways such as MAPK, NF-κB, and PI3K/AKT by binding to pattern recognition receptors such as Receptor for Advanced Glycosylation End-products (RAGE) and Toll-like Receptor 4 (TLR4) to modulate the proliferation, invasion, and microenvironmental remodeling of tumor cells, including malignancy, metastasis, and drug resistance, metastasis and microenvironment remodeling, and other malignant biological behaviors [7]. Notably, the HMGB1-mediated signaling network is involved in regulating tumor cell-intrinsic properties. It plays a pivotal role in immune escape and treatment resistance of OS by affecting immune cell function and angiogenesis. This finding provides an important theoretical basis for developing targeted therapeutic strategies based on intervention with the HMGB1 signaling axis [8].

The main objective of this review is to synthesize current knowledge on the role of HMGB1 in OS, specifically focusing on the latest progress in targeted therapies. To fulfill this aim, we conducted a literature search across multiple databases—including PubMed, Web of Science, CNKI, and Embase—using keywords such as “HMGB1,” “osteosarcoma,” “signaling pathways,” and “targeted therapy.” We prioritized both Chinese and English publications from the last five years. In addition to summarizing advances in HMGB1-targeted therapies, this review emphasizes the molecular characteristics and regulatory mechanisms of HMGB1 in OS, supporting the development of precision diagnosis and treatment systems and advancing translational medical research.

## 2. HMGB1: From Structural Features to Tumorigenic Signaling

### 2.1. Molecular Structure of HMGB1

Highly migratory group proteins (HMGB) are a large family of non-histone chromatin proteins that are ubiquitous, abundant, and highly conserved throughout the body. HMGB1 consists of a long molecule of 214 amino acids containing two homologous DNA-binding domains: the A box and the B box. The A box spans amino acids 9 to 79, and the B box spans 88 to 162. The A box and B box domains are responsible for DNA binding and bending, and there is an acidic tail spanning amino acids 186–214, with the acidic carboxyl-terminal region composed of 30 consecutive glutamic acid and aspartic acid residues [9]. The two HMG boxes of HMGB1 are structurally composed of three α-helices and two rings arranged in an “L” shape. HMGB1 has two nuclear localization signals (NLSs) located in the A box (C28-44) and B box (C179-185), respectively. NLS1 contains four conserved lysine residues, and NLS2 contains five [10]. The molecular structure of HMGB1 is shown in Figure 1.

### 2.2. Biological Effects of HMGB1 in Tumors

HMGB1 plays opposite roles in cancer development and treatment. On the one hand, HMGB1 promotes tumorigenesis, and HMGB1 overproduction caused by chronic inflammatory responses is associated with tumorigenesis. Yang et al. [11] showed that LPS induced the release of pro-inflammatory cytokines (e.g., IL-1β, IL-6, and TNF-α) in an HMGB1-dependent manner, ameliorating colon cancer progression. Furthermore, Yuan et al. [12] found that HMGB1 can be released from necrotic cells under hypoxia into the extracellular environment in growing solid tumors. Extracellular HMGB1 promotes the release of cytokines, such as IL-6 and IL-8, through activation of the MAPK- and MyD88-dependent NF-κB pathways, which stimulates the proliferation, angiogenesis, EMT, invasion, and metastasis of tumor cells. Nucleus and cytoplasmic HMGB1 promotes autophagy and inhibits apoptosis in tumor cells to induce chemoresistance. On the other hand, HMGB1 plays an important role in tumor suppression, regulating the effects of tumor radiotherapy and immunotherapy, and even in cancer prediction. For example, in the treatment of renal cell carcinoma cases, HMGB1 is thought to predict high-grade tumors, and HMGB1 differs in the cytoplasm of patients with different grades of renal cell carcinoma, with a higher distribution of HMGB1 expression being detected in the cytoplasm of patients with grade 2 or higher renal cell carcinoma compared to patients with grade 1 renal cell carcinoma [13]. HMGB1 also induces PARP1 self-modification, which promotes the interaction of PARP1 with LC3, which in turn triggers autophagy and, finally, chemotherapy resistance in small-cell lung cancer; therefore, HMGB1 may be a predictive biomarker of PARP1 response in small-cell lung cancer patients [14]. HMGB1, located in cytoplasmic lysate or mitochondria, may bind to autophagy-related genes (e.g., Beclin 1) to regulate cellular and mitochondrial autophagy. Intracellular HMGB1 acts as a tumor suppressor by binding to tumor suppressor proteins such as Rb. Xu et al. [15] experimentally demonstrated that HMGB1 could also play an antitumor role by enhancing CD8^+^ T cells. HMGB1, in the nucleus, supports the expression of IFN-γ in CD8^+^ T cells by directly regulating the activity of Eomes, a transcription factor for IFN-γ. Extracellularly, HMGB1 enhances chemotherapeutic effects by shifting tumor cells from apoptosis to senescence. In addition, HMGB1 can mediate immunogenic cell death during radiotherapy and enhance antitumor immunity. HMGB1 can be rapidly released from dead cells after chemotherapy, such as anthracyclines or radiotherapy. After release from necrotic cells or secretion by activated macrophages, HMGB1 can recruit inflammatory cells and mediate interactions between NK cells, dendritic cells, and macrophages. Activated NK cells provide an additional source of HMGB1, which is released into the immune synapse between NK cells and immature dendritic cells, promoting dendritic cell maturation and induction of Th1 responses [16]. Thus, modulating HMGB1 may provide a potential combined strategy for cancer radiotherapy and immunotherapy.

## 3. Osteosarcoma: Clinical Hallmarks and Pivotal Signaling Pathways

### 3.1. Clinical Characteristics and Therapeutic Dilemmas

As the most common primary malignant bone tumor, the high complexity of the biological behavior and clinical management of OS stems from the superposition of multidimensional factors. Regarding epidemiological characteristics, OS presents a unique bimodal age distribution, i.e., the first peak of incidence is concentrated in the child and adolescent group with rapid bone growth (median age 18 years). In comparison, the second peak is primarily seen in the elderly population over 60 years old. Notably, the two populations significantly differ in terms of pathogenesis and clinical phenotype, with adolescent OS primarily associated with osteoblasts’ malignant transformation during active skeletal growth. In contrast, elderly OS is often secondary to causative factors such as Paget’s disease [17,18]; it is worth noting that these observations are primarily based on clinical materials and cellular studies involving adolescents and elderly populations. This disparity suggests the potential significance of age stratification in the diagnosis and treatment of OS. Clinically, the onset of the disease is characterized by localized pain and swelling, which are often mistaken for growing pains and trauma in these patients, ultimately leading to delayed diagnosis [19]. Notably, those with pathological fractures caused by osteolytic destruction as the first manifestation often have tumors accidentally found due to emergency visits, highlighting the importance of clinical differential diagnosis. In terms of anatomical distribution, OS was found in the metaphysis of long bones, with the distal femur, proximal tibia, and proximal humerus being the most frequent sites [20], which may be closely related to the high metabolic state of the active bone growth region and the proliferation kinetics of osteoblasts.

However, the high metastatic tendency of OS significantly exacerbates the difficulty of clinical diagnosis and treatment. The most common condition is short-term lung metastasis, followed by distant bones and lymph nodes, and metastatic disease and local recurrence together constitute the main factors of patient death. Data suggest that approximately 30% of patients with localized lesions and 80% of patients with primary metastases eventually experience recurrence [21], a dilemma that exposes the limitations of the current treatment system in controlling micrometastases. Currently, the primary clinical treatment is neoadjuvant chemotherapy, which includes preoperative chemotherapy, intraoperative radical resection, and postoperative chemotherapy [22]. Although the neoadjuvant chemotherapy program can achieve more satisfactory therapeutic efficacy, in the process, it is simultaneously faced with multidrug resistance [23], which has inhibited a substantial breakthrough in the survival benefit for metastatic or recurrent patients, and even with active treatment, the 5-year survival rate is only 20% [2], suggesting that it is difficult to overcome this efficacy bottleneck by solely relying on traditional chemotherapeutic agents. Given the critical bottlenecks of conventional treatment options, there is an urgent clinical need to utilize new treatment strategies to improve treatment for OS patients. In addition to traditional therapeutic approaches, including surgery, chemotherapy, and radiotherapy, therapies targeting the heterogeneous molecular features of tumors and immune microenvironment modulation strategies will be the research focus. It is worth noting that the treatment of OS in the clinical setting becomes challenging due to the high level of cellular heterogeneity and the complex molecular and genetic mechanisms of development. Therefore, analyzing the source of tumor heterogeneity and developing co-targeting strategies against key molecular nodes have become important directions to overcome the current dilemma.

### 3.2. Osteosarcoma-Associated Core Signaling Pathways

A multidimensional signaling network dynamically regulates the malignant progression of OS, and its molecular mechanism presents a high degree of complexity. Among the core regulatory pathways, the PI3K/AKT pathway is pivotal to mediating chemoresistance and tumor proliferation through downstream effector molecules. In contrast, the Wnt/β-catenin pathway can synergistically drive metastatic progression through multiple pathways. Notch signaling signals abnormality and dysfunction of the Hippo pathway to form a cross-regulatory network, which promotes tumor recurrence and metastatic dissemination. At the level of microenvironmental regulation, inflammation–immunomodulation imbalance accelerates disease progression through TGF-β signaling reprogramming and aberrant expression of immune checkpoints. Despite the synergistic effects of multiple pathways revealed by these studies, OS still exhibits significant molecular heterogeneity and a lack of characteristic driver mutations, resulting in a pathogenic mechanism that has not been fully elucidated in most cases. This biological complexity has led to a stagnation in clinical efficacy over the past three decades, suggesting the need to construct integrated multidimensional genomics models, systematically analyze the microenvironment–genome interaction network, and develop co-targeting strategies based on the nodes of signaling pathways to overcome the current therapeutic bottlenecks.

Table 1 shows the relationship between different types of molecular targets and pathways and the biological function of therapeutic OS.

**Table 1 cimb-47-00887-t001:** The relationship between different types of molecular targets and biological functions for treating osteosarcoma.

Research Object	Target/Pathway	Function	Reference
ZIP10	PI3K/AKT	Promotes OS proliferation and chemotherapy resistance	[24]
MARK2	PI3K/AKT/NF-kB	Leads to resistance to cisplatin chemotherapy	[25]
EGFR	PI3K/AKT	Promotes OS progression and leads to gemcitabine resistance	[26]
Serglycin	JAK/STAT	Promotes OS proliferation, migration and invasion	[27]
miR-181a-5p	PTEN-AKT	Promotes OS progress	[28]
miR-210-5p	PIK3R5, AKT/mTOR	Promotes OS EMT and oncogenic autophagy	[29]
miR-134-5p	ITGB1/MMP2/PI3K/AKT	Inhibits malignant OS phenotype	[30]
lncRNA SNHG10	Wnt/β-catenin	Promotes proliferation and invasion of OS	[31]
USP3	PI3K/AKT	Promotes OS proliferation and metastasis	[32]
microRNA-101	PI3K/AKT, JAK/STAT	Inhibits OS tumor growth and metastasis	[33]
GABPB1-AS1	SP1/Wnt/β-catenin	Competitive binding and inhibition of miR-199a-3p promote OS progression	[34]
COL5A2	TGF-β, Wnt/β-catenin	Inhibits OS cell invasion and metastasis	[35]
CCR9	Wnt/β-catenin	Promotes OS transfer	[36]
JAG1	Notch	Promotes metastasis and recurrence of OS	[37]
YAP/TAZ	Hippo	Regulates cell proliferation and cell survival to stop organ overgrowth	[38]
ZFP36L1	SDC4-TGF-β	Inhibits OS lung metastasis	[39]
ICSBP	PD-L1	Promotes OS growth in vitro and in vivo	[40]
miR-138	DEC2	Inhibits OS proliferation and invasion	[41]
Loxl2	Wnt	Promotes invasiveness of OS	[42]
Curcumin	Nrf2/GPX4	Induces iron death and apoptosis in OS cells	[43]

## 4. Therapeutic Strategies for Osteosarcoma Targeting HMGB1 and Its Signaling Pathway

### 4.1. Advances in HMGB1 and Its Signaling Pathway in OS

#### 4.1.1. HMGB1/RAGE Pathway

RAGE is one of the most important receptors for the HMGB1 protein, belonging to the cell surface immunoglobulin superfamily. Its extracellular domain is typically divided into three structural domains: V-type, C1-type, and C2-type. The variable V domain is connected to the constant C1 and C2 domains. The integrated structure of the V domain and C1 domain primarily interacts with acidic ligands (such as HMGB1) and plays a specific role in cancer progression through multiple pathways [44]. Uncontrolled proliferation, a primary characteristic of malignant tumors, is closely associated with signaling pathways regulating cell proliferation and apoptosis. The HMGB1/RAGE pathway plays a crucial role in this process, activating downstream pathways such as Ras/MEK/ERK1/2, microRNA, NF-κB, MAPK, and PI3K/Akt, thereby regulating molecular expression levels and promoting the progression and development of cancer cells. Hou et al. [5] found that silencing HMGB1 downregulates RAGE expression levels, regulates macrophage M2 polarization, and ultimately inhibits OS cell migration and invasion. Yu et al. [45] demonstrated that circ-LRP6 downregulates miR-141-3p expression and upregulates HMGB1 expression, thereby promoting OS cell proliferation, migration, and invasion, confirming that the activation of this pathway can regulate miRNA expression and influence tumor proliferation. HMGB1/RAGE can also promote ATP production, regulate cellular metabolism, and trigger neutrophil-mediated necrosis-induced damage amplification, thereby providing a favorable environment for OS cell proliferation [46].

#### 4.1.2. HMGB1/TLR4 Pathways

Currently, 13 types of TLR (TLR1-TLR13) have been identified in mammals. Among all subtypes, TLR4, as a transmembrane pattern recognition receptor, can be activated by endogenous enzyme-mediated cascade reactions through recognizing bacterial outer-membrane lipopolysaccharides, shock proteins released during cell necrosis, and other substances participating in the body’s non-specific immune response [47]. The HMGB1/TLR4 pathway is one of the pathways that initiate inflammatory responses. Increasingly, studies have shown that the inflammatory expression of TLR4 plays an important role in regulating the inflammatory environment around tumors and tumor progression [48]. More importantly, the HMGB1/TLR4 pathway has multiple effects on cancer proliferation and invasion. Studies have shown that HMGB1/TLR4 can upregulate the expression levels of matrix metalloproteinase-9 (MMP-9). After expression, MMP-9 can degrade and disrupt the extracellular matrix on the tumor surface, enabling tumor cells to infiltrate and grow into surrounding tissues along the missing basement membrane, ultimately leading to the invasion and metastasis of OS cells. Tachibana et al. [49] found that LPS, as a ligand for TLR4, can interfere with the immunosuppressive activity of myeloid-derived suppressor cells (MDSCs) in vitro, demonstrating that HMGB1 can enhance and regulate MDSC inhibitory activity through the downstream molecule TLR4. During OS cell proliferation, HMGB1/TLR4 pathway activation may inhibit CD8^+^ T cells via downstream MDSCs, thereby enabling immune escape in OS. Therefore, TLR4 inhibitors can influence immune responses and tumor invasion, offering significant research potential for targeted therapy in OS.

#### 4.1.3. Integrin-Dependent Signaling Pathway

Integrins αvβ3 and αvβ5 play a crucial role in OS, serving not only as mediators of various cell adhesion- and migration-related signaling pathways but also being closely associated with interactions with HMGB1 [50]. In a study on the oncolytic adenovirus OBP-502, Mochizuki et al. [51] found that OBP-502 targets integrins αvβ3 and αvβ5 through the arginine–glycine–aspartic acid peptide sequence on its fiber protrusions, thereby specifically infecting and lysing tumor cells. This targeting mechanism enhances the virus’s selective infectivity toward tumor cells and activates antitumor immune responses by inducing immunogenic cell death. Specifically, after infecting tumor cells, OBP-502 releases damage-related molecular patterns such as HMGB1 and adenosine triphosphate (ATP), which activate the immune system and promote antitumor immune responses. Additionally, OBP-502 infection significantly upregulates the expression of programmed cell death ligand one on the surface of tumor cells. This upregulation may be associated with tumor cells’ sensitivity to immune checkpoint inhibitors, suggesting that the integrin-dependent signaling pathway also plays an important role in immune regulation. Although this study did not directly investigate the detailed molecular mechanisms of integrin-dependent signaling pathways in OS, the results suggest that integrins αvβ3 and αvβ5 may play a potential role in the invasion and immune evasion of OS. By targeting these integrins, new therapeutic strategies can be developed to enhance the efficacy of immunotherapy and inhibit tumor progression.

#### 4.1.4. Non-Coding RNA Regulatory Network

Non-coding RNA (ncRNA) plays a crucial role in OS through exosome-mediated intercellular communication, target gene regulation, and signal pathway interaction, forming a complex regulatory network in OS proliferation, metastasis, and drug resistance. Recent studies have shown that miRNAs, lncRNAs, and circRNAs dynamically regulate immune responses and metabolic reprogramming in the tumor microenvironment through synergistic or antagonistic actions. These make them important targets for precision therapy in OS. Exosomes secreted by OS cells remodel the tumor microenvironment by delivering functional ncRNAs. For example, studies indicate that long non-coding RNAs within exosomes, such as RP11-361F15.2, can promote OS progression by regulating the polarization of M2-type tumor-associated macrophages. Concurrently, LIFR-AS1 within exosomes can also modulate miR-29a, thereby influencing the proliferation and invasion of OS cells [52,53]. Additionally, circPRKD3 reprograms the immune microenvironment by inhibiting the STAT3 signaling pathway and recruiting CD8^+^ T cells, suggesting that exosomal ncRNAs possess dual functions in tumor promotion and immune regulation [54]. circ_0005909 binds to miR-936 to upregulate HMGB1, promoting epithelial–mesenchymal transition and tumor metastasis [55]. Zhou et al. [6] reported that the long non-coding RNA Lnc-PHF3-3 is overexpressed in OS doxorubicin-resistant cell lines and patient tissues. It competitively binds to miR-142-3p to release its inhibition on HMGB1, thereby upregulating HMGB1 expression. Mechanistic studies revealed that Lnc-PHF3-3 binds to the Ago2 protein to form an RNA-induced silencing complex, specifically enriching miR-142-3p and inhibiting its function. Dual-luciferase reporter assays further confirmed that miR-142-3p directly targets the 3′UTR region of HMGB1, while knockdown of Lnc-PHF3-3 significantly reduced HMGB1 protein levels and reversed doxorubicin resistance. Additionally, in a mouse model overexpressing Lnc-PHF3-3, HMGB1-mediated autophagy activation and apoptosis inhibition were significantly enhanced, reducing tumor responsiveness to chemotherapy. These findings reveal the core role of the Lnc-PHF3-3/miR-142-3p/HMGB1 axis in OS resistance and provide new targets for targeting the non-coding RNA-HMGB1 regulatory network. HMGB1 and its associated signaling pathways are presented in Figure 2.

### 4.2. Therapeutic Strategies Targeting HMGB1 and Its Signaling Pathways

#### 4.2.1. Inhibition of HMGB1 Expression in OS

Previous studies have shown that patients with osteosarcoma (OS) who have high HMGB1 expression tend to have lower survival rates and a poorer prognosis. This effect is stronger when HMGB1 moves from the nucleus or mitochondria to the cytoplasm. This process may increase its oncogenic effects. Blocking HMGB1 expression, particularly preventing its changes in subcellular localization, may offer a potential therapeutic strategy [51,56,57,58]. Yi et al. [59] found that the inhibitory effects of Gynostemma pentaphyllum saponins on the malignant biological behavior of OS cells may be related to the inhibition of the HMGB1-RAGE signaling pathway, suggesting that ginsenosides may serve as targeted therapeutic agents for OS. Additionally, recent studies have highlighted that microRNAs (miRNAs) act as regulators of HMGB1. Zhang et al. [60] found that miR-129-5p is downregulated in OS cells, attenuating their proliferation and migration capabilities while reducing HMGB1 expression, suggesting that miR-129-5p may exert a tumor-suppressing role in OS. Li et al. [61] found that overexpression of miR-505 and HMGB1 silencing inhibited OS cell proliferation, migration, and invasion while increasing apoptosis rates. Co-transfection of miR-505 and si-HMGB1 exhibited a more significant inhibitory effect on OS cell proliferation and invasion, with a higher apoptosis rate, suggesting that miR-505 may inhibit OS cell proliferation and invasion by targeting and inhibiting HMGB1 while promoting apoptosis. Lv et al. [62] found that miR-1284 functions by directly binding to the 3′-UTR of HMGB1 and regulating its expression. Overexpression of miR-1284 inhibited cell proliferation and migration while altering the protein expression of genes associated with epithelial–mesenchymal transition (EMT), suggesting that miR-1284 is a negative regulator of HMGB1, capable of inhibiting OS cell proliferation and migration. Liu et al. [63] conducted dual-luciferase reporter gene analysis, RT-qPCR, and Western blot analysis, confirming that miR-935 can directly target the 3′-untranslated region of HMGB1 and negatively regulate HMGB1 expression in OS cells. Further studies demonstrated that miR-935 inhibits OS cell proliferation and invasion by directly targeting HMGB1. Guo et al. [64] found that miR-22 inhibits OS cell proliferation and migration by targeting HMGB1 and suppressing HMGB1-mediated autophagy. These studies suggest that miRNAs play an oncogene-suppressing role in OS and may serve as candidate molecules for targeted therapy. Huang et al. [65] pointed out that OS chemotherapy resistance induced by KLF4 (Krüppel-like factor 4) may be mediated by upregulating HMGB1 expression, suggesting that targeting the KLF4/HMGB1 pathway may be a potential therapeutic strategy to overcome OS chemotherapy resistance. Liu et al. [66] demonstrated that a novel androstenedione derivative, DSTD, enhances OS chemotherapy sensitivity by inhibiting the expression of macrophage migration inhibitory factor in MG-63 and U2OS cells, thereby reducing HMGB1 levels.

Although numerous studies have demonstrated that inhibiting HMGB1 expression may offer a new therapeutic strategy for OS, this field of research still faces significant challenges. First, experimental data indicate that various miRNAs (miR-129-5p, miR-505, and miRNA-1284, etc.), ginsenosides, and KLF4 can effectively suppress HMGB1 expression under in vitro conditions, significantly inhibit tumor cell proliferation and migration, and induce apoptosis. However, the in vivo translation of these experimental results remains unclear and requires further validation through more in-depth experimental studies. Second, current research primarily focuses on the mechanisms of action of individual miRNAs, while the efficacy and safety of combining multiple miRNAs have not been systematically evaluated. Additionally, miRNAs, as biological small molecules, exhibit poor stability and low targeting specificity, necessitating the development of efficient delivery carriers (such as liposomes or exosomes) for targeted precision therapy. Improving these miRNAs’ delivery efficiency and reducing potential side effects are also pressing issues. Furthermore, whether substances such as Gynostemma pentaphyllum saponins and DSTD may cause severe adverse reactions in patients requires further evidence-based medical research in future studies. Based on this, future research should explore feasible pathways for translating these therapeutic strategies from basic research to clinical practice. In the future, the development of combination therapies, such as the combination of miRNAs with chemotherapy/immunotherapy (e.g., miR-505 + cisplatin) or the synergistic use of natural compounds with targeted drugs (e.g., Gynostemma pentaphyllum saponins + HMGB1 inhibitors), may also represent promising treatment options.

#### 4.2.2. Target Downstream Signaling Pathways of HMGB1 in OS

Many studies have identified numerous new target sites in the downstream pathways mediated by HMGB1. Yan et al. [67] found that psoralen isoflavones inhibit cell viability, proliferation, migration, and invasion by regulating the HMGB1-mediated p38 MAPK signaling pathway but increase cell apoptosis in a concentration-dependent manner. Wang et al. [67] found that magnoflorine significantly inhibited HMGB1 expression and NF-κB activation but upregulated miR-410-3p levels. HMGB1 overexpression promoted NF-κB activation and reversed the effects of magnoflorine on OS cell viability, invasion, EMT, and cisplatin sensitivity, suggesting that it inhibits the malignant phenotype of OS cells and increases cisplatin sensitivity by regulating the miR-410-3p/HMGB1/NF-κB pathway. The above studies suggest that psoralen isoflavones and magnoflorine may be novel therapeutic agents for OS. Wang et al. [68] reported that low levels of miR-122-5p and miR-204-5p, along with HMGB1 overexpression, can reduce the inhibitory effects of circLRP6 on OS cell proliferation, migration, and invasion. Additionally, circ-LRP6 can promote OS progression by regulating the miR-141-3p/HDAC4/HMGB1 axis [45]; Zhang et al. [67] found that circBBS9 promoted OS progression by regulating the miR-485-3p/HMGB1 axis, and circBBS9 knockdown weakened OS growth in vivo. These findings suggest that circBBS9 may serve as a prognostic biomarker and therapeutic candidate for OS patients. Ding et al. [55] found that OS patients with high circ_0005909 expression had lower survival rates. Inhibition of circ_0005909 reduced tumor growth in vivo and restricted cell viability, colony formation, migration, invasion, and EMT in vitro, suggesting that it inhibits OS progression by downregulating HMGB1 via miR-936. Luo et al. [69] found that HNF1A-AS1 blocks OS progression through the miR-32-5p/HMGB1 axis, providing potential therapeutic targets and prognostic biomarkers for OS patients. Liu et al. [70] found that knocking down MALAT1 reduces HMGB1 expression, inhibits OS cell growth, and promotes apoptosis. Inhibitors of miR-142-3p and miR-129-5p partially restored the inhibitory effects of MALAT1 knockdown on HMGB1 expression, OS cell growth, and apoptosis promotion, and these results suggest that the miR-142-3p/miR-129-5p/HMGB1 axis promotes OS progression. Zhou et al. [6] identified a novel lncRNA named Lnc-PHF3-3, which promotes chemoresistance to azithromycin in OS cells via the miR-142-3p/HMGB1 axis, suggesting that inhibiting Lnc-PHF3-3 could serve as a therapeutic strategy. Li et al. [71] found that LncRNA HULC induces OS progression by regulating the miR-372-3p/HMGB1 signaling axis. This suggests that targeting LncRNA is a promising therapeutic strategy for OS patients.

Although studies on HMGB1-mediated downstream signaling pathways have provided multiple potential strategies for OS treatment, such as inhibiting tumor progression by regulating pathways including p38 MAPK and miR-410-3p/HMGB1/NF-κB, the clinical application of these findings still faces significant obstacles. The precise regulatory mechanisms of circular RNAs (e.g., circ_LRP6, circBBS9) and long non-coding RNAs (e.g., Lnc-PHF3-3, HULC) on HMGB1 and its associated pathways remain poorly understood. Although preliminary evidence suggests that these molecules can modulate OS cell behavior, the complexity and diversity of their functions require further investigation to assess their feasibility as diagnostic biomarkers and therapeutic targets. Given the high heterogeneity of OS, an intervention targeting a single pathway may be insufficient to inhibit tumor progression completely. Exploring multi-pathway combination therapies may become a key strategy to enhance treatment efficacy. As our understanding of HMGB1 and its downstream pathways deepens, complex interactions between different signaling pathways are gradually emerging. For example, certain circRNAs and lncRNAs not only regulate HMGB1 expression but may also indirectly influence the tumor microenvironment or immune response through unknown mechanisms. Therefore, elucidating these multilevel regulatory networks is crucial for developing effective therapeutic strategies. Additionally, constructing animal models that more accurately reflect human pathophysiological characteristics is essential for evaluating the efficacy and safety of potential therapeutic methods, which can help reduce the risk of clinical trial failure and accelerate the drug development process. Furthermore, developing efficient drug delivery systems to ensure that biomolecules can precisely and safely target the lesion site and exert their functions is a critical step in achieving clinical translation. Finally, in the era of highly developed precision medicine, personalized medicine, which considers individual patient differences’ influence on treatment responses, is very important. Future research should focus on developing personalized treatment regimens targeting HMGB1 based on multidimensional data such as genomics and transcriptomics to achieve optimal therapeutic outcomes.

The different therapeutic approaches targeting the expression of HMGB1 and its downstream signaling pathways are summarized in Table 2 and Figure 3.

**Table 2 cimb-47-00887-t002:** Overview of therapeutic approaches targeting HMGB1 and its downstream signaling pathways.

Research Object	Target/Pathway	Function	Reference
miR-505	HMGB1	Inhibits OS cell proliferation and invasion and promotes cell apoptosis	[61]
Gynostemma pentaphyllum saponins	HMGB1-RAGE	Inhibits malignant biological behavior of OS cells	[59]
Psoralen isoflavone	p38 MAPK/HMGB1	Inhibits OS cell migration and invasion	[67]
Magnoflorine	miR-410-3p/HMGB1/NF-κB	Inhibits the malignant phenotype of OS cells and increases sensitivity to cisplatin	[72]
miR-1284	HMGB1	Inhibits OS cell proliferation and migration	[62]
miR-129-5p	HMGB1	Inhibits OS cell proliferation and migration and promotes cell apoptosis	[60]
miR-935	HMGB1	Inhibits OS cell proliferation and migration	[63]
miR-22	HMGB1	Inhibits OS cell proliferation and migration	[64]
KLF4	HMGB1	Leads to OS chemotherapy resistance	[65]
DSTD	HMGB1	Enhances OS chemotherapy sensitivity	[66]
circLRP6	miR-141-3p/HDAC4/HMGB1	Promotes OS progress	[45]
circBBS9	miR-485-3p/HMGB1	Promotes OS progress	[73]
circ_0005909	miR-936/HMGB1	Inhibits OS progress	[55]
HNF1A-AS1	miR-32-5p/HMGB1	Inhibits OS progress	[69]
MALAT1	miR-142-3p/miR-129-5p/HMGB1	Promotes OS progress	[70]
Lnc-PHF3-3	miR-142-3p/HMGB1	Leads to OS chemotherapy resistance	[6]
HULC	miR-372-3p/HMGB1	Promotes OS progress	[71]

In summary, targeting HMGB1 and its downstream pathways offers a multidimensional strategy for OS treatment. However, challenges such as the complexity of mechanisms, delivery efficiency, and translational model limitations must be overcome. Future research should focus on multi-target synergistic interventions, developing precise delivery systems, and optimizing preclinical models to accelerate the transition from basic research to clinical application.

## 5. Conclusions

HMGB1, as a highly regarded potential therapeutic target, has demonstrated significant research value in sepsis, autoimmune diseases, and various cancers. In sepsis, although HMGB1 levels in patient serum are markedly elevated, its association with clinical prognosis and specific clinical value remain unclear [74]. In the field of autoimmune diseases, anti-HMGB1 antibodies have demonstrated efficacy in animal models such as rheumatoid arthritis [75]. However, their safety profile, immunological characteristics, and clinical applicability require further evaluation. Currently, a relevant clinical trial for systemic lupus erythematosus (NCT05193591, registered at ClinicalTrials.gov) is underway.

In cancer research, HMGB1 overexpression has been demonstrated to be closely associated with the development of multiple tumor types [76]. For instance, HMGB1 plays a significant role in predicting treatment response in adult-onset juvenile rhabdomyosarcoma [77]. In clinical trials involving patients with pancreatic cancer, inhibiting HMGB1 has been shown to enhance sensitivity to chemotherapy [78]. Overall, HMGB1-targeted therapy shows potential across various cancer fields, including colorectal cancer and non-small-cell lung cancer (NCT01312467, NCT02186847, registered at ClinicalTrials.gov). Although dedicated clinical trials for osteosarcoma remain limited, existing experimental data provide a theoretical basis for targeting HMGB1 in the treatment of osteosarcoma. As more clinical studies progress, the potential of HMGB1 as a therapeutic target for cancer warrants further validation.

With the deepening of research in molecular oncology and precision medicine, the central regulatory role of HMGB1 and its mediated signaling network in the occurrence, development, and treatment resistance of OS has been widely recognized. Numerous essential and clinical studies have demonstrated that HMGB1 binds to pattern recognition receptors such as RAGE and TLR4, activating downstream pro-inflammatory and pro-survival signaling pathways, including NF-κB and MAPK. This directly drives tumor cell proliferation, invasion, and metastasis and reshapes the immune microenvironment by regulating non-coding RNA networks and mediating treatment resistance. In recent years, intervention strategies targeting the HMGB1 pathway have shown a trend toward diversification, including gene regulation based on non-coding RNA (miRNA, circRNA, lncRNA), natural compound screening, and small-molecule inhibitor development. These strategies have all demonstrated promising antitumor effects in preclinical studies.

However, the translation of HMGB1-targeted therapy into clinical practice still faces three key challenges: (1) functional complexity: HMGB1 exhibits dual functions of promoting tumorigenesis and regulating antitumor immunity, and its effects are temporally and spatially regulated; (2) technical bottlenecks: RNA therapies inherently suffer from poor stability and low delivery efficiency; (3) tumor heterogeneity: the high genomic instability of OS limits the efficacy of single-target strategies. These challenges suggest that future research should focus on ① developing multi-target synergistic intervention strategies, ② integrating multi-omics data for precise subtyping, and ③ establishing a more comprehensive preclinical evaluation system.

Looking ahead, the following research directions are particularly important: (1) further elucidating the dynamic regulatory mechanisms of HMGB1 in the spatiotemporal heterogeneity of OS; (2) developing disease models closer to clinical settings, such as organoids and humanized mice; (3) innovating delivery technologies such as nanocarriers to enhance treatment specificity; and (4) integrating imaging omics, single-cell sequencing, and artificial intelligence to construct an “imaging-molecular” multimodal diagnosis and treatment system. Through interdisciplinary collaborative innovation, targeted combined therapeutic strategies targeting HMGB1 hold promise for overcoming current treatment challenges in OS and opening new avenues for improving patient outcomes.

## Figures and Tables

**Figure 1 cimb-47-00887-f001:**
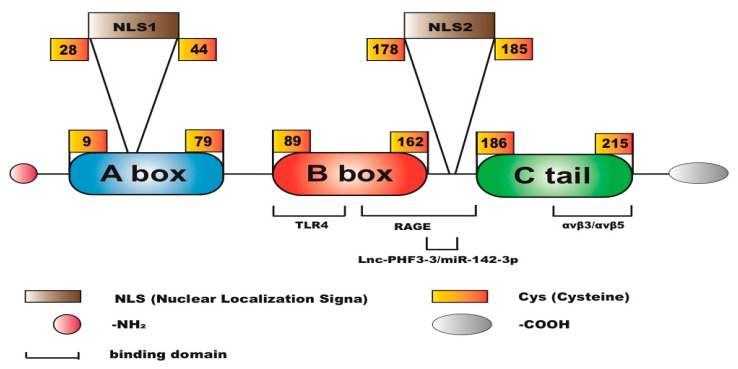
Molecular structure of HMGB1.

**Figure 2 cimb-47-00887-f002:**
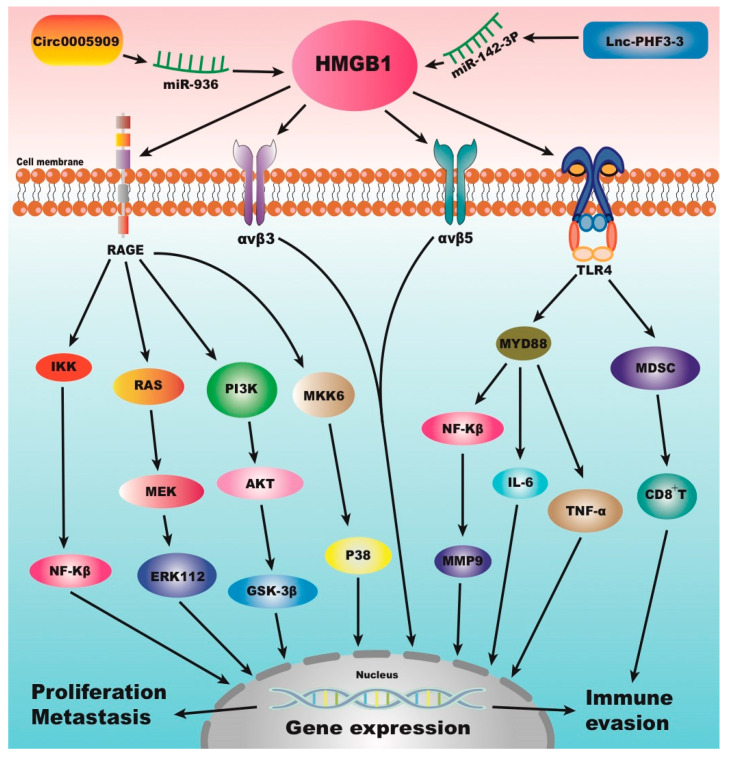
HMGB1 and its signaling pathways.

**Figure 3 cimb-47-00887-f003:**
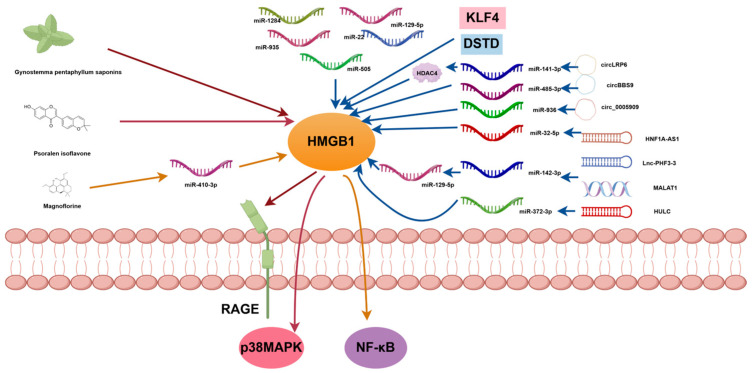
Schematic diagram of therapeutic methods targeting HMGB1 and its downstream signaling pathways.

## Data Availability

No new data were created or analyzed in this study. Data sharing is not applicable to this article.

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
