# Peer review of "HMGB1 and Its Signaling Pathway in Osteosarcoma: Current Advances in Targeted Therapy"

_cimb, 2025, doi:10.3390/cimb47110887_

Round 1
Reviewer 1 Report
Comments and Suggestions for Authors
This manuscript entitled “HMGB1 and Its Signaling Pathway in Osteosarcoma: Current Advances in Targeted Therapy” is submitted by Dr Liu, Dr Ye, and their colleagues in Huaqiao University.
In this review article, the authors summarize the current knowledge about HMGB1, especially in osteosarcoma. HMGB1 is a well-studied protein in the research fields of infection, inflammation, and tissue damage/repair. The authors focused on summarizing the biological function, signal pathway regulation mechanism, and intervention strategy of HMGB1 in osteosarcoma to explain the applicability of HMGB1-targeting agents in osteosarcoma treatment. The authors organized the reports into 4 sections: 1. Introduction 2. HMGB1: From structural features to tumorigenic signaling 3. Osteosarcoma: Clinical hallmarks and pivotal signaling pathways 4. Therapeutic strategies for osteosarcoma targeting HMGB1 and its signaling pathway. However, the linkages among the sections seem needed to be further improved.
Here are some suggestions and comments on the manuscript:
-
- On page 3… the title of 2.2 is the same as “2.1 Molecular structure of HMGB1”.
- In the research field of anti-cancer therapy development, descriptions about the physiological roles of the target proteins in normal tissue and of the cancer specificity are quite important. In this paper, the authors appear to be missing the clinical and translational results that support the explanation of why targeting HMGB1 is feasible and useful for treating OS.
- HMGB1 is a protein with multiple functions dependent on its subcellular localization. Therefore, it is suggested that the explanation of the subcellular localization of HMGB1 in OS cells is informative in understanding the oncogenic roles of HMGB1 and the methods for targeting HMGB1 in cancer treatment. Ref. 50 is not enough to describe that.
- Clinical trials targeting High Mobility Group Box 1 (HMGB1) are underway or being developed for inflammatory diseases, sepsis, and certain cancers, using strategies like anti-HMGB1 antibodies and other antagonists. Whether these regimens are potentially useful or under clinical trial of OS treatment seems able to be summarized.
- Ref. 42 does not appear to support the author's description. The report published on Theranostics (Apr 8;14(6):2605-2621) in 2024 seems more suitable.
- To Ref. 46, it will be better replaced by the other correlated reports (doi: 10.1016/j.canlet.2019.12.041; doi: 10.1186/s12935-021-01893-0).
Author Response
Comments 1: On page 3… the title of 2.2 is the same as “2.1 Molecular structure of HMGB1”.
Response 1: Thank you for your valuable feedback. As per your suggestion, we have revised the title of section 2.2 to "Biological effects of HMGB1 in tumours" to eliminate the duplication with section 2.1. We believe this change enhances the clarity and organization of the manuscript.
The revised content is as follows: (Page 3, Line 80)
Comments 2: In the research field of anti-cancer therapy development, descriptions about the physiological roles of the target proteins in normal tissue and of the cancer specificity are quite important. In this paper, the authors appear to be missing the clinical and translational results that support the explanation of why targeting HMGB1 is feasible and useful for treating OS.
Response 2: Thank you for your thoughtful and insightful comment. We agree that describing the physiological roles of target proteins in normal tissues and their cancer specificity is crucial for advancing anti-cancer therapies. We acknowledge that clinical data on targeting HMGB1 is still limited, with most studies primarily relying on animal models, particularly in vitro and in vivo experiments. In our manuscript, we have already included a significant amount of experimental data supporting the feasibility of targeting HMGB1 for OS treatment, and we have provided relevant summaries in various sections. Additionally, we conducted a search on several clinical trial databases, including ClinicalTrials.gov, but unfortunately, we were unable to find any ongoing or completed clinical trials specifically focused on OS treatment with HMGB1-targeting therapies. Given the current stage of research, we believe that the existing experimental data is sufficiently compelling to support the potential of targeting HMGB1 for OS therapy. While we fully acknowledge the importance of clinical trials, we hope the current focus on the relevant experimental data is deemed appropriate at this stage. We sincerely appreciate your suggestion and have highlighted the changes in red for your convenience.
The revised content is as follows: (Page 9, Lines 329-350), (Page 11, Lines 409-413)
Comments 3: HMGB1 is a protein with multiple functions dependent on its subcellular localization. Therefore, it is suggested that the explanation of the subcellular localization of HMGB1 in OS cells is informative in understanding the oncogenic roles of HMGB1 and the methods for targeting HMGB1 in cancer treatment. Ref. 50 is not enough to describe that.
Response 3: Thank you for your valuable suggestion. In response to your comment, we have added three additional references to further support the explanation of the subcellular localization of HMGB1 in osteosarcoma cells. These references (doi:10.1007/s00262-020-02774-7, doi:10.3892/ol.2015.3907, doi:10.1016/j.jconrel.2025.113614) have been highlighted in red in the revised manuscript for your convenience. We believe these additions enhance the manuscript by providing a more comprehensive understanding of the oncogenic roles of HMGB1 and its potential as a therapeutic target in osteosarcoma.
The revised content is as follows: (Page 8, Lines 292-297)
Comments 4: Clinical trials targeting High Mobility Group Box 1 (HMGB1) are underway or being developed for inflammatory diseases, sepsis, and certain cancers, using strategies like anti-HMGB1 antibodies and other antagonists. Whether these regimens are potentially useful or under clinical trial of OS treatment seems able to be summarized.
Response 4: Thank you for your insightful suggestion. We agree that summarizing the potential clinical relevance of HMGB1-targeting therapies for OS is important. While clinical trials targeting HMGB1 are currently underway for inflammatory diseases, sepsis, and certain cancers, clinical trials specifically focused on osteosarcoma are still limited. In response to your comment, we have added a brief summary of these ongoing trials and their potential relevance to OS treatment, acknowledging that further research is needed. The changes have been highlighted in red for your convenience.
The revised content is as follows: (Page 12, Lines 428-447)
Comments 5: Ref. 42 does not appear to support the author's description. The report published on Theranostics (Apr 8;14(6):2605-2621) in 2024 seems more suitable.
Response 5: Thank you for your helpful suggestion. In response to your comment, we have replaced the previous reference with the one you recommended, doi:10.7150/thno.92672 , which we believe provides more robust and relevant support for the content. This reference offers valuable insights and strengthens our argument. The change has been reflected in the revised manuscript, and the updated reference has been highlighted in red for your convenience.
The revised content is as follows: (Page 6, Lines 221-224)
Comments 6: To Ref. 46, it will be better replaced by the other correlated reports (doi: 10.1016/j.canlet.2019.12.041; doi: 10.1186/s12935-021-01893-0).
Response 6: Thank you for your constructive feedback. In response to your comment, we have cited the two recommended articles, doi:10.1186/s12935-021-01893-0 and doi:10.1016/j.canlet.2019.12.041 , and have made corresponding revisions to the manuscript to better align with these references. The changes have been highlighted in red for your convenience.
The revised content is as follows: (Page 7, Lines 226-270)
Reviewer 2 Report
Comments and Suggestions for Authors
Osteosarcomas are among the least well-studied tumors, despite being among the most common tumors of mesenchymal origin. Certain research groups devote considerable attention to the study of osteosarcomas, and therefore periodic systematization of accumulated knowledge in reviews is essential.
The review is well structured and the outline is excellent, but a more in-depth description of the molecular events associated with HMGB1 expression, the regulation of its expression, and the impact of changes in its expression on tumorigenesis and tumor response to therapy is essential. The authors should rely on the analysis of experimental articles rather than reviews or other summary sources. Furthermore, in Chapter 3.1, the authors note that osteosarcomas in young people (children and adolescents) and the elderly develop through different mechanisms. However, nowhere further in the description of the biological processes associated with HMGB1 it is stated whether the study was conducted on cells or clinical material characteristic of "young" osteosarcomas or "elderly " osteosarcomas.
There are also several specific comments.
Abstract. In the Abstract, describe how the articles for the review were found and which databases were used. For example, the search was conducted using the keywords "osteosarcoma" and "therapy" in PubMed.
Introduction. In the Introduction, as in the Abstract, it is advisable to describe how the literature search was conducted (which databases, libraries, or other collections, and which keywords). Were only English-language sources considered, or were publications in different languages ​​used? Was the search conducted over recent years or over the entire period of osteosarcoma research?
Introduction. The same sentence is repeated twice in the second paragraph of the Introduction.
The Introduction contains no references.
The last paragraph of the Introduction contains a strange phrase. Explain what "OS.sion" means.
Figure 1. The title of Figure 1 does not correspond to what is shown in the figure. The figure shows a schematic structure of the HMGB1 protein.
Page 3. In the middle of paragraph 2.2. Molecular structure of HMGB1, there is the sentence "HMGB1 deficiency leads to defective autophagy and increased apoptosis, which leads to tumorigenesis." This is a somewhat odd statement. Tumor formation is typically associated with decreased apoptosis. This sentence requires more detailed explanation.
Page 4. The middle of the second paragraph. Again, one phrase is repeated twice.
Table 1. The line COL5A2 contains a hieroglyph, making the text impossible to understand for readers without knowledge of the relevant language.
Author Response
Comments 1: The review is well structured and the outline is excellent, but a more in-depth description of the molecular events associated with HMGB1 expression, the regulation of its expression, and the impact of changes in its expression on tumorigenesis and tumor response to therapy is essential. The authors should rely on the analysis of experimental articles rather than reviews or other summary sources. Furthermore, in Chapter 3.1, the authors note that osteosarcomas in young people (children and adolescents) and the elderly develop through different mechanisms. However, nowhere further in the description of the biological processes associated with HMGB1 it is stated whether the study was conducted on cells or clinical material characteristic of "young" osteosarcomas or "elderly " osteosarcomas.
Response 1: Thank you for your thoughtful and constructive feedback. We appreciate your positive remarks regarding the structure and outline of the review. In response to your comment, we have expanded the description of the molecular events associated with HMGB1 expression, its regulation, and the impact of changes in its expression on tumorigenesis and tumor response to therapy. As per your suggestion, we have placed greater emphasis on experimental articles, which can be found in the reference section at the end of the manuscript, to provide a more in-depth and reliable analysis. Additionally, regarding Chapter 3.1, we have clarified that the studies discussed focus on osteosarcomas in both younger and older populations. We have now specified whether the studies were conducted on cell lines or clinical materials characteristic of "young" osteosarcomas or "elderly" osteosarcomas. These additions and clarifications have been incorporated into the manuscript and highlighted in red for your convenience.
The revised content is as follows: (Page 14, Line 524), (Page 4, Lines 130-133)
Comments 2: Abstract. In the Abstract, describe how the articles for the review were found and which databases were used. For example, the search was conducted using the keywords "osteosarcoma" and "therapy" in PubMed.
Response 2: Thank you for your suggestion. In response to your comment, we have revised the abstract to include a description of how the articles for the review were found and which databases were used. This change has been reflected in the revised abstract and highlighted in red for your convenience.
The revised content is as follows: (Page 1, Lines 11-16)
Comments 3: Introduction. In the Introduction, as in the Abstract, it is advisable to describe how the literature search was conducted (which databases, libraries, or other collections, and which keywords). Were only English-language sources considered, or were publications in different languages used? Was the search conducted over recent years or over the entire period of osteosarcoma research?
Response 3: Thank you for your valuable suggestion. In response to your comment, we have revised the Introduction to include a description of how the literature search was conducted, including the databases, libraries, and keywords used. We considered both English and Chinese-language sources, with a significant proportion of the literature coming from the past five years. These changes have been incorporated into the revised manuscript and highlighted in red for your convenience.
The revised content is as follows: (Page 2, Lines 54-63)
Comments 4: Introduction. The same sentence is repeated twice in the second paragraph of the Introduction.
Response 4: Thank you for pointing that out. We have carefully reviewed the second paragraph of the Introduction and removed the repeated sentence. The revised manuscript reflects this change, and it has been highlighted in red for your convenience.
The revised content is as follows: (Page 1, Line 39)
Comments 5: The Introduction contains no references.
Response 5: Thank you for your observation. In response to your comment, we have added eight relevant references to the Introduction to support the background and context of the review. These additions have been highlighted in red in the revised manuscript for your convenience.
The revised content is as follows: (Page 14, Lines 525-542)
Comments 6: The last paragraph of the Introduction contains a strange phrase. Explain what "OS.sion" means.
Response 6: Thank you for pointing that out. We apologize for the confusion. The phrase "OS.sion" was a typographical error. We have corrected it, and in response to your previous comment (Comments 3), we have revised the entire paragraph to include more detailed information regarding the literature search and the inclusion of relevant references. The changes have been highlighted in red for your convenience.
The revised content is as follows: (Page 2, Lines 54-63)
Comments 7: Figure 1. The title of Figure 1 does not correspond to what is shown in the figure. The figure shows a schematic structure of the HMGB1 protein.
Response 7: Thank you for your valuable comment. In response to your observation, we have updated the title of Figure 1 to "Figure 1. Molecular structure of HMGB1" to accurately reflect its content. The change has been highlighted in red for your convenience.
The revised content is as follows: (Page 3, Line 79)
Comments 8: Page 3. In the middle of paragraph 2.2. Molecular structure of HMGB1, there is the sentence "HMGB1 deficiency leads to defective autophagy and increased apoptosis, which leads to tumorigenesis." This is a somewhat odd statement. Tumor formation is typically associated with decreased apoptosis. This sentence requires more detailed explanation.
Response 8: Thank you for your insightful comment. After carefully considering your feedback, we have decided to remove the sentence in question. We recognize that it could have led to confusion, and we apologize for any misunderstanding this may have caused. The revised manuscript now omits this sentence, and the change has been highlighted in red for your convenience.
The revised content is as follows: (Page 3, Line 104)
Comments 9: Page 4. The middle of the second paragraph. Again, one phrase is repeated twice.
Response 9: Thank you for your careful review. We have carefully checked the second paragraph on Page 4 and removed the repeated phrase. The change has been reflected in the revised manuscript and highlighted in red for your convenience.
The revised content is as follows: (Page 4, Lines 155-157)
Comments 10: Table 1. The line COL5A2 contains a hieroglyph, making the text impossible to understand for readers without knowledge of the relevant language.
Response 10: Thank you for pointing that out. We apologize for the inclusion of the hieroglyph in Table 1. We have corrected the issue by replacing it with the appropriate characters. The change has been reflected in the revised manuscript and highlighted in red for your convenience.
The revised content is as follows: (Page 5, Line 188)
Round 2
Reviewer 2 Report
Comments and Suggestions for Authors
I thank the authors for their attention to the comments. The authors responded to all comments and made appropriate revisions to the manuscript. The review is ready for publication in its current form.
Author Response
Dear Reviewer,
Thank you for your positive feedback and for confirming that our manuscript is now ready for publication. We sincerely appreciate your time and valuable comments, which have greatly improved our work.
Sincerely,
Zhuosheng Liu
Email: 13799221029@163.com